# Identification of *Rf* Genes in Hexaploid Wheat (*Triticum*
*aestivum* L.) by RNA-Seq and Paralog Analyses

**DOI:** 10.3390/ijms22179146

**Published:** 2021-08-24

**Authors:** Mirosław Tyrka, Beata Bakera, Magdalena Szeliga, Magdalena Święcicka, Paweł Krajewski, Monika Mokrzycka, Monika Rakoczy-Trojanowska

**Affiliations:** 1Department of Biotechnology and Bioinformatics, Rzeszów University of Technology, Powstańców Warszawy 12, 35-959 Rzeszów, Poland; mtyrka@prz.edu.pl (M.T.); mszeliga@prz.edu.pl (M.S.); 2Department of Plant Genetics, Breeding and Biotechnology, Warsaw University of Life Sciences, Nowoursynowska 166, 02-787 Warszawa, Poland; b.bakera@biol.uw.edu.pl (B.B.); magdalena_swiecicka@sggw.edu.pl (M.Ś.); 3Institute of Plant Genetics, Polish Academy of Sciences, Strzeszyńska 34, 60-479 Poznań, Poland; pkra@igr.poznan.pl (P.K.); mmok@igr.poznan.pl (M.M.)

**Keywords:** fertility restoration, gene expression, mapping, microspore, pentatricopeptide repeat genes, pollen

## Abstract

Among the natural mechanisms used for wheat hybrid breeding, the most desirable is the system combining the cytoplasmic male sterility (cms) of the female parent with the fertility-restoring genes (*Rf*) of the male parent. The objective of this study was to identify *Rf* candidate genes in the wheat genome on the basis of transcriptome sequencing (RNA-seq) and paralog analysis data. Total RNA was isolated from the anthers of two fertility-restorer (Primépi and Patras) and two non-restorer (Astoria and Grana) varieties at the tetrad and late uninucleate microspore stages. Of 36,912 differentially expressed genes (DEGs), 21 encoding domains in known fertility-restoring proteins were selected. To enrich the pool of *Rf* candidates, 52 paralogs (PAGs) of the 21 selected DEGs were included in the analyses. The expression profiles of most of the DEGs and PAGs determined bioinformatically were as expected (i.e., they were overexpressed in at least one fertility-restorer variety). However, these results were only partially consistent with the quantitative real-time PCR data. The DEG and PAG promoters included *cis*-regulatory elements common among PPR-encoding genes. On the basis of the obtained results, we designated seven genes as *Rf* candidate genes, six of which were identified for the first time in this study.

## 1. Introduction

Common wheat (*Triticum aestivum* L.) is one of the most important cereals cultivated worldwide because it is a staple food crop [1,2,3]. Despite its importance, in recent years, there has been a relatively slow increase in global wheat production (https://www.statista.com/statistics/267268/production-of-wheat-worldwide-since-1990/; http://www.fao.org/3/y3557e/y3557e08.htm, accessed on 16 August 2021). The development of hybrid cultivars may increase wheat yields by 10% [4,5,6]. Nevertheless, hybrid wheat currently represents only a small proportion of commercially produced wheat. Common wheat undergoes self-pollination, usually involving cleistogamous florets [7,8,9]. The key challenges for breeding wheat hybrids are as follows: modulating floral development and architecture to enable outcrossing, regulating male sterility, and restoring fertility [10,11].

Different techniques and natural mechanisms may be used to prevent self-pollination and facilitate the production of hybrid wheat. Chemical hybridizing agents (CHAs) that selectively alter pollen shedding and viability are useful for emasculating wheat in trials conducted to estimate germplasm combining ability and produce commercial hybrid seeds [10,12,13]. Among the natural mechanisms that have been used for wheat hybrid breeding, the most desirable system in practice combines the cytoplasmic male sterility (*cms*) of the female parent with the fertility-restoring genes (dominant *Rf* alleles) of the male parent [10,14]. The cytoplasm of *Triticum timopheevii* is the most frequently used cytoplasm for sterilizing wheat [14,15] because it induces male sterility without affecting female fertility and it has relatively few deleterious effects [16]. This *cms* system, which may be the most effective, has serious limitations that are mainly associated with an extremely low frequency of *Rf* alleles and the substantial complexity of the fertility restoration, which requires two or three of the nine identified major restorer loci [14,17]. Other reported limiting factors of the *T. timopheevii cms* system include the instability of male sterility [18,19,20] and the strict requirements for temperature and day-length [21,22].

Most *Rf* genes encode pentatricopeptide repeat (PPR) proteins, but there are a few exceptions, including the rice *Rf2* and *Rf17* genes, respectively, encoding a mitochondrial glycine-rich protein [23] and a 178-amino acid mitochondrial sorting protein containing an acyl-carrier protein synthase-like domain [24] or the maize *Rf2* (*Rf2a*) gene encoding the aldehyde dehydrogenase domain (ADD) [25,26]. The RF proteins prevent the accumulation of products encoded by mitochondrial *cms*-conferring open reading frames [27,28].

To date, eight fertility restorer genes (*Rf1–Rf6* and *Rf8–Rf9*) that restore the production of normal pollen in wheat with the *T. timopheevii* sterilizing cytoplasm have been identified and mapped on the following seven wheat chromosomes: 1A (*Rf1*), 1BS (*Rf3*), 2DS (*Rf8*), 6A (*Rf6*), 6AS (*Rf9*), 6B (*Rf6* and *Rf4*), 6D (*Rf5*), 7B (*Rf7*), and 7D (*Rf2*) [29,30,31,32,33,34,35,36,37,38,39]. There is some uncertainty regarding the nomenclature of *Rf2*, *Rf3*, *Rf4*, and *Rf6* [40]. Moreover, the location of *Rf6* remains undetermined, with some researchers indicating this gene is on chromosomes 6A and 6B (e.g., [37,38]), whereas others have stated this gene is on chromosome 5D (e.g., [41,42]).

Most *Rf* genes were characterized in terms of the strength of their effect, and some interactions between these genes have been reported. The *Rf3* gene, which was first identified on chromosome 1B of European spelt [31], is one of the most effective genes for restoring male fertility in common wheat [40,43]. The donors of dominant *Rf3* alleles include wheat cultivar Primépi, which also contains *Rf6* [41], and the spelt cultivars Badenkrone, Badenstern, and Schwabenspelz [31,40,43]. The *Rf3* gene was reported as dominant, but it is suppressed by a weak fertility suppressor gene on chromosome 7D [31,39,43].

Both *Rf1* and *Rf4* were described as major, codominant restorer genes [30,33,43] affected by several recessive modifiers influenced by the environment [44,45]. Geyer et al. [46] determined that *Rf1* is more highly expressed than *Rf4* and that the effects of these loci are not additive. The *Rf2* and *Rf5* genes were considered to be minor restorer genes [33]. The major restorer gene *Rf6* was transferred to wheat from chromosome 6U of *Aegilops umbellulata* Zhuk. [35]. As described above, its location in the wheat genome is unknown.

The dominance of *Rf8* localized on chromosome 2DS was verified by Sinha et al. [36]. Another major dominant restorer gene, *Rf9*, was detected in the subtelomeric region of chromosome 6AS [39]. Melonek et al. [47] recently identified *Rf1* and *Rf3* via genetic mapping, comparative sequence analyses, and functional studies. The proteins encoded by these two genes bind to the mitochondrial *orf279* transcript of a newly identified mitochondrial gene and induce its cleavage, thereby blocking the expression of the *cms* trait. They observed that *Rf1* and *Rf3* have no effect on another mitochondrial gene, *orf256*, previously considered to cause male sterility in wheat. Two other rye-derived genes restoring male fertility, *Rfc3* (on 6RL) and *Rfc4* (on 4RL), were detected in wheat–rye addition lines [48]. These restorer genes, particularly *Rfc3*, may be useful for hybrid wheat and triticale breeding programs.

Several quantitative trait loci (QTLs) for general fertility restoration were also mapped on wheat chromosomes 1AS, 1BS, 2DS, 4BS, 6AS, 6BS, 7AL, and 7D at positions overlapping *Rf* genes [39,46,49,50,51,52]. Additionally, several fertility-related QTLs were localized to regions that were not reported to carry *Rf* genes (i.e., 1DS, 2AL, 2BS, 4AL, 4BS, 5A, 5BL, and 7AL) [39,49,50,52].

Other than fertility restorer genes effective against G-type cytoplasm, one dominant restorer gene, *Rfv1*, and several QTLs effective against *Aegilops kotschyi* cytoplasm (S^v^-type) were identified on the short arm of chromosome 1B [49,53,54]. The mapped position of *Rfv1* is very close (2 cM) to *Rf3* [55]. Four other genes, *Rfm1* [56], *Rfn1* [57], *Rfk1* [58], and *Rfu1* [59], encode fertility restorers effective against the cytoplasm of *Aegilops mutica* (T-type), *Aegilops uniaristata* (N-type, previously designated as M^u^), *Ae. kotschyi*, and *Ae. umbellulata*, respectively. All of these genes are localized on chromosome 1B, but the first three genes are on the short arm. Tsunewaki [60] detected three genes (*Rfv1*, *Rfm1*, and *Rfn1*) in the same 2.9-cM subsegment bordered by the S-6 and *Xucr*-*5* markers on the 1BS arm in Pavon and nine other examined wheat cultivars. The locus was named *Rf*-^multi^ (i.e., restoration of fertility in multiple CMS systems). Additionally, a major restorer gene (*Rfd1*) effective against *Aegilops crassa* cytoplasm was identified on the long arm of chromosome 7B [61].

Because of the development of next-generation sequencing (NGS) technology, structural data obtained from chromosome engineering and genetic mapping approaches can now be coupled with functional characteristics for gene discovery. Transcriptome sequencing (RNA-seq) is a powerful and cost-efficient tool for studying transcriptomes. It has been widely used to investigate model and non-model organisms to identify and quantify RNA, profile transcripts, detect single nucleotide polymorphisms (SNPs), and identify genes differentially expressed between samples [62].

The objective of this study was to identify and characterize *Rf* candidate genes in two fertility-restoring wheat genotypes (Primépi and Patras) compared to the non-restoring cultivars (Astoria and Grana). The RNA-seq data were first processed generally and then used to specifically identify *Rf* candidate genes and their paralogs (PAGs) and analyze gene expression profiles during pollen development.

## 2. Results

### 2.1. RNA-Seq Analysis

#### 2.1.1. General Characteristics of the RNA-Seq Data

The sequencing of 24 libraries resulted in 861.3 million raw reads, which included 860.5 million high-quality clean reads. The number of clean reads across the 24 RNA-seq libraries ranged from 29.4 to 44.3 million. The transcriptomes of fertility-restoring and non-restoring cultivars were compared at the tetrad (T) and at late uninucleate microspore (M) stages. The number of upregulated and downregulated DEGs varied among the 12 comparisons. The Grana vs. Patras and Patras vs. Primépi transcriptome comparisons at the M stage resulted in the most upregulated and downregulated DEGs. The fewest upregulated and downregulated DEGs were detected in the Grana vs. Primépi transcriptome comparison at the M stage (Table 1).

A total of 36,912 DEGs were identified between the transcriptomes of the two fertility-restoring cultivars and the two non-fertility-restoring cultivars at two anther developmental stages. In this pool, 20,368 DEGs were not redundant, and 9173 (45%) were unique to comparisons C1–C12 (Table 1). At the T stage, 2842, 2329, 243, and 19 DEGs were common to two, three, four, and five comparisons, respectively. This analysis revealed 14 unique DEGs that distinguished Primépi from the non-fertility-restoring cultivars. Similarly, 126 unique DEGs were useful for distinguishing Patras from the non-fertility-restoring cultivars at the T stage (Figure 1a).

At the M stage, 3420, 2216, 119, and 5 DEGs were common to two, three, four, and five comparisons, respectively. Two genes (*TraesCS7A02G439300* and *TraesCS7B02G338200*) were identified as DEGs in all six comparisons. In this developmental stage, a single DEG was specific to Primépi (C5 and C6), whereas 69 DEGs were specific to Patras (C7 and C8), implying these genes are useful for distinguishing these cultivars from the non-fertility-restoring cultivars (Figure 1b). The DEG analysis indicated 769 of the 36,912 DEGs, including *Rf* candidates, may be useful for distinguishing the fertility-restorers from the non-fertility-restorers. Moreover, 210 restorer-specific DEGs were identified in both developmental stages.

#### 2.1.2. Targeted Analysis of RNA-Seq Results

The general analysis detected many genes differentially expressed in the fertility-restoring cultivars. To identify the genes related to fertility restoration, the local database of proteins corresponding to *Rf* genes (Appendix A) was compared with the *T. aestivum* reference genome, which resulted in the selection of 120,744 wheat genes. These genes were narrowed down to 535 genes with a sequence similarity and sequence coverage exceeding 49%. The 535 genes were compared with the 20,368 non-redundant DEGs identified in comparisons C1–C12 (Table 1). This enabled the selection of 40 DEGs present in both pools, representing 0.2% of the initial number of DEGs. Finally, an E-value < 0.05 was applied as a criterion for selecting 21 DEGs to be analyzed in greater detail (Table 2, Appendix A). With the exception of six DEGs (DEG7, DEG8, DEG14, DEG16, DEG17, and DEG19), which were unique to a specific comparison (most of them were detected in the Astoria vs. Patras comparison at the T stage), the DEGs were detected in more than one comparison (up to six). The expression levels of most of the DEGs revealed by multiple comparisons were stably upregulated or downregulated in the fertility-restoring cultivars. The exceptions were DEG3, DEG4, DEG5, and DEG15; the expression levels of these DEGs were upregulated or downregulated in the fertility restorers depending on the comparison.

Each of the finally selected 21 DEGs matched 1–16 reference sequences. Fourteen DEGs (DEG1–DEG14) encoded proteins that were similar to at least three reference sequences (*Brachypodium distachyon* protein Rf1, mitochondrial (XP_024315805.1); *Zea mays* PPR-814a (ACN24620.1); and *Aegilops tauschii* subsp. *tauschii* protein Rf1, mitochondrial-like (XP_020153657.1)) and included the consensus PPR motif. All DEGs in this group encoded proteins with PPR and tetratricopeptide (TPR)-like helical domains and were similar to RNA-binding proteins belonging to the mitochondrial group I intron splicing factor CCM1 family (Table 2). These DEGs were localized on chromosomes 1A, 1B, 2A, 2B, 6A, and 6B. Generally comprising one to three exons, the DEGs represented genetically differentiated groups with three loci with low allelic variation (1–9 alleles), four moderately variable loci (20–85 alleles), and seven highly variable loci (132–266 alleles). According to a BLAST search, the remaining seven DEGs (DEG15–DEG21) only matched *Z. mays Rf2* (NP_001105891.1) and encoded the same ADD. These sequences were mapped to chromosome 6B and the homoeologous group 7 chromosomes; the genes in this group comprised 7–11 exons. Single splicing variants were reported for the DEGs, with the exception of *TraesCS7B02G116800*, which had two splicing variants. The seven DEGs encoding the ADD were characterized by high allelic variation, with 38–290 alleles (mean of 211 alleles) (Table 2).

In silico expression patterns of the selected 21 DEGs, which were determined on the basis of the NGS read counts, revealed the qualitative and quantitative differences in the expression levels and patterns (Figure 2). The expression pattern of PPR genes was genotype- and gene-specific. Very low expression levels were found for DEG1, DEG2, DEG3, DEG7, DEG8, and DEG13 compared to the other genes. Among the DEGs encoding the PPR domain (DEGs 1–14), eight were clearly overexpressed in the fertility-restoring cultivar, Patras. The expression levels of two DEGs, DEG9 and DEG14, were higher in the second fertility-restoring cultivar, Primépi (but only in the T stage for DEG14). The expression levels of the four remaining DEGs in the PPR family were higher in one or both non-fertility-restoring cultivars (usually Astoria). The expression levels and patterns of the DEGs encoding the ADD exhibited greater genotype and developmental stage specificity. For example, DEG15 and DEG18 were more highly expressed in Patras than in both non-fertility-restoring cultivars in the T and M stages, respectively. The DEG19 expression level was higher in Primépi than in Astoria and Grana, but only in the M stage. The remaining DEGs encoding the ADD were usually overexpressed in the non-fertility-restoring cultivars.

### 2.2. Analysis of Paralogous Genes

In order to increase the pool of *Rf* candidate genes, the reference genome was analyzed to screen for PAGs of the 21 selected DEGs. A total of 1165 PAGs were identified using the Ensembl Plants database. This set included 21 previously selected DEGs, which were paralogs of each other. Next, 775 PAGs with a mean expression level in the samples exceeding 10 were used for the unsigned weighted correlation network analysis, which revealed 19 clusters M1, M2, …, M19 comprising genes with correlated expression patterns (Figure 3). Among these clusters, only one (M8) comprised 10 PAGs (PAG7 belonged to the ADD family, whereas the other PAGs belonged to the PPR family) that were expressed differently in both fertility-restorers than in the two non-fertility restorers in both pollen developmental stages (Figure 3).

Generally, the expression patterns of individual genes from cluster M8 were similar to the normalized expression pattern, with expression higher in fertility restorers (Figure 4), but there were some clear differences. For example, PAG7, PAG8, and PAG9 were more highly expressed in the non-fertility-restorers than in the fertility restorers.

Most of the PAGs in clusters M9 (10 genes) and M17 (11 genes), which were all from the PPR family (Appendix A), had average expression levels that were higher in the fertility-restoring cultivar Patras than in the non-fertility-restoring cultivars Astoria and Grana in both stages. The expression profiles of individual PAGs from these two groups were mostly consistent with the normalized patterns (although in some cases (e.g., PAG13 or PAG24), the expression level differences between the restorer and non-restorer cultivars were not significant). With the exception of PAG20 in cluster M9 as well as PAG23, PAG24, and PAG27 in cluster M17, all PAGs were more highly expressed in Primépi than in the other cultivars, including Patras. The largest cluster (M18) comprised 21 genes that were expressed at higher levels in the restorer cultivars than in Astoria. In this cluster, PAG50 was the same as one of the 21 DEGs selected according to the RNA-seq data (i.e., DEG16) (Appendix A). An analysis of individual PAGs in this group indicated that the expression levels of most of the PAGs were consistent with the normalized profile. The six exceptions were PAG32, PAG34, PAG38, PAG44, PAG48, and PAG51, which were more highly expressed in Astoria than in the other three cultivars. Three other PAGs were expressed at higher levels in both restorers than in both non-restorers, but only in one developmental stage (PAG40 and PAG52 in the M stage and PAG45 in the T stage). However, if compared only with Astoria, the expression levels of most PAGs were higher in both fertility-restoring cultivars in the T and M stages, which is similar to the normalized profile. In cluster M18, three PAGs (PAG41, PAG44, and PAG50) belonged to the ADD family, whereas the remaining 18 PAGs were members of the PPR family. The expression patterns of individual PAGs from clusters M9, M17, and M18 are presented in Appendix A.

The expression patterns of the PAGs in the other seven clusters (M3, M5, M6, M7, M10, M12, and M19) were less specific. The PAGs more highly expressed in Patras than in the other cultivars in the first stage formed clusters M3, M5, and M7. The PAGs in clusters M6 and M10 were most highly expressed in Primépi in the T stage, whereas the PAGs in clusters M12 and M19 were most highly expressed in Primépi in the M stage. Regarding the remaining clusters (M1, M2, M4, M11, M13, M14, M15, and M16), the average expression patterns were usually genotype-specific (Figure 3).

Fifty-two PAGs (including PAG50, which was the same as DEG16) from clusters M8, M9, M17, and M18 were chosen for additional analyses.

### 2.3. In Silico Analysis of the Promoters of Selected DEGs and PAGs

To better characterize the selected pool of DEGs and PAGs, a bioinformatic analysis of their promoters was performed. The 2000-nt sequences upstream of the 21 selected DEGs included 89 *cre* (Appendix A). In addition to the common *cre* (CAAT-box and TATA-box), some identified *cre* were over-represented (i.e., detected more than 50 times, with a mean frequency exceeding 2.4 per DEG). Among the over-represented *cre* were those related to disease resistance and stress responses (e.g., MYB-binding site, MYB-like sequence, and MYC), plant development, secondary metabolism and signal transduction (MYC-binding site), signal transduction and hormone responses (CGTCA-motif, TGACG-motif, and ABRE), light responses (G-box), and those with unknown functions (STRE, WRE3, and AS-1). The most common *cre* were the MYBs, with a frequency greater than eight per 100 nt. Of the 12 rarest *cre* in the DEG promoters, five (Box III, chs-CMA2a, chs-Unit 1 m1, Pc-CMA2c, and Gap-box) were related to light responses, three (AT-rich element, AT-rich sequence, and TGA-box) were associated with hormone responses, two (HD-Zip 1 and HD-Zip 3) influenced developmental processes, one (AC-II) was related to vascular tissue-specific expression, and one (JARE) was not functionally characterized (Appendix A). Three *cre* (AT-rich sequence, Box II-like sequence, and Box III) were detected only once in the whole DEG set (Appendix A). The most frequent *cre* (at least seven per 100 nt) were detected in DEG1, DEG3, and DEG21, whereas the least frequent *cre* (1.9 per 100 nt) were detected in DEG7 (Appendix A). The mean frequency of the *cre* in the promoters of DEGs encoding the PPR domain was almost identical to the mean frequency of the *cre* in the promoters of DEGs encoding the ADD (Appendix A).

A similar examination of the 2000-nt sequences upstream of 52 PAGs revealed 103 *cre*, which was 13 more than the number of *cre* detected in the DEG promoters. The 13 *cre* unique to the PAG promoter set were as follows: 3-AF3-binding site, AAAC-motif, AACA-motif, CAG-motif, chs-CMA2b, F-box, E2Fb, GC-repeat, L-box, motif I, NON, OCT, and Pc-CMA2a (Appendix A).

Excluding the common *cre* (CAAT-box and TATA-box) in the PAG promoters, the most frequent *cre* (i.e., detected more than 135 times, with a mean frequency exceeding 2.60 per PAG), were the same as the most frequent *cre* among the DEGs, with the exception of WRE3 (unknown function). The frequency of the common MYB *cre* was only slightly lower for the PAGs (7.77 per 100 nt) than for the DEGs. Of the 17 least frequent *cre*, which were usually detected in only one PAG promoter, six (AAAC motif, chs-CMA2b, L-box, LS7, Pc-CMA2a, and Pc-CMA2c) were associated with light responses and five were related to specific processes, including endosperm-specific negative expression (AACA motif), vascular tissue-specific expression (AC-II), auxin responses (AuxRE), development (HD-Zip 3), and root-specific gene regulation (motif I). Additionally, the 3-AF3-binding site was included in a conserved DNA module array. The functions of the remaining five *cre* (ACTCATCCT sequence, AT1-motif, CAG-motif, F-box, and OCT) remain unknown. The PAGs with the highest *cre* frequencies were PAG12, PAG44, and PAG46, whereas PAG37 was revealed as the PAG with the lowest *cre* frequency (2.55 per 2000 nt) (Appendix A). Two rarely occurring *cre* (AuxRE and HD-Zip 3) were detected in the 2000-nt sequences upstream of the DEGs and PAGs (Appendix A).

A comparison among the promoters of DEGs, PAGs, and 12 reference genes generally did not reveal any major common features. Four *cre* (4cl-CMA1b, H-box, telo-box, and Y-box) were unique to the reference genes. The most frequent *cre* were similar in all three gene groups. Moreover, among the most frequent *cre*, five (ABRE, CGTCA-motif, TGACG-motif, STRE, and AS-1; the first three are related to signal transduction pathways) were common to the reference genes, DEGs, and PAGs. The MYBs occurred less frequently, but they were still relatively common (5.75 per 100 nt). In terms of the rarest *cre*, they were most commonly detected in the reference genes (Appendix A).

### 2.4. Mapping DEGs and PAGs in Rf Gene Regions

The next step was the mapping of 21 DEGs identified during the targeted RNA-seq analysis (Table 2), and the 52 detected PAGs (including PAG 50, which is the same as DEG16) (Appendix A) were mapped on the wheat reference genome (IWGSC RefSeq v1.0) along with the available genetic markers linked to known fertility restoration genes (Appendix A). Moreover, the physical positions of 1690 PPR domain-encoding genes in the CS genome (version 1) available in the URGI database (*TaPPR* genes) and 62 loci encoding proteins with the ADD (Appendix A) were used to saturate regions comprising *Rf* genes and QTLs.

The seven physical regions corresponding to *Rf* gene positions, two additional fertility-related loci on 1B, and three QTLs mapped highly precisely on 2A, 5B, and 7A were used to identify co-localized DEGs and PAGs (Figure 5, Appendix A). Five DEGs were localized on chromosomes 1A and 1B within regions with previously mapped *Rf1* and *R3* genes. The remaining 16 DEGs were assigned to regions that did not overlap with reported *Rf* loci. Genetic mapping data were sufficient for identifying physical regions corresponding to *Rf1* (1A: 5.02–32.9 Mb), *Rf3* (1B: 16.85–44.07 Mb), *Rf4* (6B: 3.88–35.52 Mb), *Rf6* (5D: 371.47–445.11 Mb), *Rf8* (2D: 13.75–23.02 Mb), and *Rf9* (6A: 0.96–23.72 Mb). An anchor marker (*Xgdm130*) for *Rf2* was also physically mapped to 7D (4.2 Mb), but there were no genetic markers for *Rf5* and *Rf7*. A detailed analysis of the markers mapped to chromosome 1B revealed two major loci related to fertility restoration in addition to *Rf3*. The first locus (*QRf.Sv-1B*), which explained 13.9%–17.6% of the seed set variation, was delimited to the 336.21–363.15 Mb region. The second region (1B: 506.33–557.12 Mb), which explained 16.8%–38.2% of the seed set variation, was previously reported as *Rf3* (Ahmed et al., 2001, Zhou et al., 2005). The physical mapping of fertility restoration QTLs resulted in the detection of a region on 2A (705.51–724.30 Mb), which explained 8.3%–21.0% of the seed set variation and minor loci on 5B (6.65–16.42 cM) and 7A (674.28–674.80 cM). The remaining QTLs were localized to large physical regions (1D, 2B, and 4A) or were mapped with single markers (4B and 5A). Partial physical maps revealed the co-localization of DEG1 and DEG5 with the recently mapped *Rf1* and *Rf3* genes (Figure 5).

Of the 52 PAGs, only three (PAG16, PAG17, and PAG34) were physically mapped to regions affecting fertility. Specifically, PAG34 was mapped to the same region as the QTL *QRf.Sv-1B* (336.2–336.4 Mbp). Additionally, PAG17 was mapped to a position (18.8 Mb) in the *Rf4* region delimited to 3.9–35.5 Mbp (Ma and Sorrels, 1995, Geyer et al., 2018). Finally, PAG16 was mapped to a position (14.36 Mb) in the *Rf9* region (5.4–23.7 Mbp) flanked by the markers IWB33595 and IWB73288 [39]. Both PAGs mapped in the *Rf4* and *Rf9* regions were more highly expressed in Patras than in Astoria and Grana in the M stage.

### 2.5. qRT-PCR Analysis of Selected DEGs and PAGs

To validate the in silico expression data, a quantitative reverse transcription PCR (qRT-PCR) analysis of 11 DEGs and 11 PAGs, all encoding the PPR domain, was performed. For 10 DEGs (DEG4, DEG5, and DEG7–DEG14) and 10 PAGs (PAG1, PAG5, PAG10, PAG15–PAG17, PAG22, PAG23, PAG27, and PAG31), the main selection criterion was the expected expression profile determined in silico (i.e., DEGs and PAGs that were more highly expressed in at least one fertility-restoring cultivar than in at least one non-fertility-restoring cultivar). Both DEG6 and PAG34 were chosen because they were expressed at much higher levels in the non-fertility-restoring cultivar Astoria. The presence of at least one *cre* in the 5′ upstream regulatory sequence was an additional criterion.

Regarding the DEGs, in most cases, the gene expression analyses produced relatively consistent data (Figure 2 and Figure 6). In contrast, the expression levels of two DEGs (DEG9 and DEG14) differed substantially between the in silico and qRT-PCR analyses. More specifically, according to the in silico analysis, DEG9 and DEG14 were most highly expressed in Primépi, particularly in the T stage, whereas the qRT-PCR data indicated these genes were most highly expressed in Patras in the T stage. There were some inconsistencies in the expression analyses of DEG4, DEG5, and DEG8. The DEG4 and DEG5 expression profiles were consistent in the M stage. The DEG8 expression pattern determined by qRT-PCR was consistent with the results of the in silico data in the T stage, but in the M stage, the qRT-PCR analysis indicated this gene was more highly expressed in Patras only when compared with the corresponding Grana expression level. The highest consistency between the qRT-PCR and in silico data was observed for DEG3, which was most highly expressed in Astoria.

In contrast to the DEGs, the qRT-PCR data for the PAGs were consistent with the in silico data for only three PAGs (PAG1, PAG10, and PAG31) (Figure 6, Appendix A). The qRT-PCR expression data for PAG5, PAG15, and PAG34 were partially consistent with the results of the in silico analysis, especially in the T stage and in the comparison between Patras and Astoria. However, the expression profiles varied considerably between the two expression analyses for PAG16, PAG22, PAG23, and PAG27. The biggest discrepancy was observed for PAG22 and PAG27. The expression profiles of these PAGs in both fertility-restoring cultivars were the complete opposite in the qRT-PCR and in silico assays.

### 2.6. Annotation of PPR Genes

To identify the genes most likely responsible for fertility restoration, 14 DEGs and 48 PAGs in the CS reference genome that were initially assigned to the PPR family (Appendix A) were characterized in terms of their encoded PPR domain structures. Eight DEGs (DEG1, DEG2, DEG4–DEG7, DEG9, and DEG11) were predicted to encode P-class PPR proteins belonging to the RFL subclade. The remaining DEGs (DEG3, DEG8, DEG10, and DEG12–DEG14) encoded proteins with 15–20 P-sites.

On the basis of their encoded PPR domains, 7 PAGs (PAG1, PAG5, PAG10, PAG16, PAG25, PAG32, and PAG52) were classified in the RFL subclass, whereas another 20 PAGs (PAG8, PAG9, PAG11, PAG13, PAG15, PAG18, PAG19, PAG22, PAG27, PAG33–PAG38, PAG42, PAG43, PAG45, PAG47, and PAG48) were assigned to the P-class, but they contained fewer than 15 tandem repeats. The remaining 21 PAGs (PAG2–PAG4, PAG6, PAG12, PAG14, PAG17, PAG20, PAG21, PAG23, PAG24, PAG26, PAG28–PAG31, PAG39, PAG40, PAG46, PAG49, and PAG51) belonged to the PLS-class.

## 3. Discussion

Modern plant cultivars are expected to have high-value agronomic and nutritional properties. Moreover, protecting the intellectual property rights of cultivar breeders is becoming an increasingly important issue. Therefore, the breeding of hybrid cultivars is warranted because such cultivars are usually associated with increased yield and quality. They also protect the interests of breeders because growers must purchase the F_1_ seeds from licensed distributors every time, owing to an inability of the growers to reproduce the seeds themselves. The seeds of all currently registered wheat F_1_ cultivars were produced according to relatively efficient and uncomplicated methods involving CHAs. Unfortunately, in some European countries, including Poland, the application of CHAs for hybrid seed production is forbidden. Furthermore, the disadvantages of using CHAs include the need for dosage optimization, relatively high costs, the poor performance of CHA-based hybrids, and poor seed germination [10,14]. Accordingly, wheat hybrid breeding methods that exploit natural mechanisms, especially those related to sterilizing cytoplasm and *Rf* genes, must be developed. Major limitations of such methods, however, are the complexity of the fertility restoration process mediated by nine *Rf* genes, of which two or three major restorer loci are required to fully restore fertility and the extremely low frequency of dominant *Rf* alleles. 

In this study, we aimed to identify and characterize wheat *Rf* genes by performing a transcriptome analysis followed by untargeted and targeted analyses of the generated data. Over the past decade, RNA-seq has become indispensable for transcriptome-wide analyses of DEGs [63]. Several studies have used RNA-seq to identify candidate genes and QTLs controlling specific traits and processes in wheat [64,65,66,67,68]. However, the utility of RNA-seq for identifying *Rf* genes has not been explored. Moreover, RNA-seq is demanding in terms of sequencing data coverage and computational analysis, but it generates gene expression data that can be further compared in different combinations. Because of its sufficient sensitivity and sequencing depth, RNA-seq is more suitable and affordable for comparative gene expression studies than microarrays, and it generates more data, including for genes that are expressed at low levels [69,70].

For the transcriptome analyses in the current study, RNA was extracted from anthers at the T and M developmental stages. These stages were selected on the basis of numerous preliminary experiments as well as literature searches. Ye et al. [71], looking for molecular mechanisms responsible for male sterility/fertility conversion in wheat with *Ae. kotschyi* thermosensitive cytoplasmic male sterility selected three anther developmental stages (i.e., late uninucleate, binucleate, and trinucleate) for their RNA-seq analysis. Comparing anther development under sterile and fertile conditions, they detected the most significant differences at the late uninucleate stage. Additionally, most genes related to male sterility/fertility conversion were differentially expressed at the late uninucleate stage, which is one of the stages we analyzed. Ye et al. [71] did not investigate the tetrad stage (i.e., the T stage in our study) because they did not detect any obvious differences in anther development between sterile and fertile conditions. However, one should consider the essential differences between both systems of sterility and, consequently, the differences in the expression of genes mediating fertility restoration. Furthermore, the mitochondrial gene *orf279*, which is important for the *cms* trait [47], is highly expressed in the early anthesis stage. Its transcript, which is targeted by the proteins encoded by at least two *Rf* genes (*Rf1* and *Rf3*), is also produced at a relatively early microsporogenesis stage. Thus, searching for *Rf* candidate genes at this stage makes sense. Earlier studies on the fertility restoration genes of other cereal species, namely *Rf2* and *Rf4* in rice and *Rf4* in maize, revealed that these genes are overexpressed during meiosis and in the uninucleate microspore (and in binucleate and trinucleate microspores in rice) pollen developmental stages [13,23,72].

The RNA-seq analysis of 24 libraries in this study detected 20,368 DEGs (among 860.5 million high-quality reads). The efficiency of this transcriptome sequencing analysis was higher than that of earlier studies, including a study by Xiao et al. [73] that determined the transcriptional changes in the roots of low-cadmium-accumulating winter wheat under cadmium stress, a study by Xiong et al. [74], who used RNA-seq to search for candidate genes associated with salinity tolerance, and a study by Iquebal et al. [66], who identified drought-responsive molecular pathways in wheat roots by analyzing the transcriptome.

To precisely filter candidate *Rf* genes from such a large number of reads, a multi-stage selection procedure was applied. Finally, 21 DEGs were designated as candidate fertility restorer genes to be further analyzed. The selected DEGs were at least 50% similar to one or more sequences in a local database comprising the sequences of all known *Rf*, PPR domain-encoding, and ADD-encoding genes in the Poaceae species. Their potential role as fertility restorers was verified following a BLAST analysis using known *Rf* genes as queries. Two-thirds of these DEGs matched PPR domain-encoding genes, whereas the remaining DEGs matched the maize *ZmRf2* (*Rf2a*) gene, which encodes the ADD. The genes involved in restoring the fertility of wheat containing the *T. timopheevii* sterilizing cytoplasm that have been described to date encode regulatory proteins (e.g., RNA splicing factors) belonging to the P-class PPR family. These proteins stabilize organellar transcripts and intron splicing [75,76,77]. A recent study by Melonek et al. [47] proved that two PPR proteins (encoded by *Rf1* and *Rf3*) target the mitochondrial *orf279* transcript. The binding of these two proteins to this transcript leads to cleavage and prevents the expression of the *cms* trait. The maize *Rf2a* gene is considered to be the only *Rf* gene encoding a mitochondrial aldehyde dehydrogenase that restores male fertility to plants with CMS-T [78]. However, the results of our current study suggest that at least some wheat *Rf* candidate genes can encode proteins with the ADD. The target of the protein encoded by the maize *Rf2* gene in this class remains unknown, but RF2A most likely oxidizes a broad range of aldehydes [67].

The in silico expression profiles for most of the selected DEGs were as expected (i.e., they were overexpressed in at least one restorer cultivar and in at least one developmental stage), especially the DEGs encoding the PPR domain in Patras. Surprisingly, only four DEGs (DEG9 and DEG14 belonging to the PPR family and DEG19 and DEG20 encoding the ADD) were overexpressed in Primépi, which is considered to be an *Rf* gene donor.

To increase the pool of *Rf* candidate genes, 21 DEGs were subjected to a PAG analysis, which resulted in the identification of 775 PAGs in 19 clusters. However, only one cluster (M8) comprised PAGs that were differentially expressed between the fertility-restoring cultivars and the non-fertility-restoring cultivars in the M and T stages. Clusters M9, M17, and M18 comprised PAGs that were either restorer cultivar-specific (M9 and M17) or developmental stage-specific (M18). The PAGs in the remaining 15 clusters were unlikely to be fertility restorer genes. This supplementary, highly effective analysis revealed 51 new *Rf* candidate genes, most of which belonged to the P-class PPR family, implying they encode RNA splicing factors.

A bioinformatics analysis of the promoters of the DEGs, PAGs, and 12 reference genes enabled the identification of dozens of *cre*. Substantial similarities were observed between the DEG, PAG, and reference gene promoters, both in terms of *cre* frequency and type. Notably, among the most frequent *cre* that were present in almost all reference gene, DEG, and PAG promoters, ABRE, the CGTCA-motif, and the TGACG-motif are associated with signal transduction pathways involved in stress responses. Chen et al. [79] (2018) detected ABRE and the TGACG motif in a 1.5-kb sequence upstream of 491 rice PPR-encoding genes. The ABRE identified by Chen et al. [79] was also detected in the gene promoters analyzed in this study, albeit at a slightly lower frequency. These findings support the hypothesis formulated by Chen et al. [79] that the genes belonging to the PPR family are responsive to stress. We propose that DEGs encoding the ADD are also related to stress responses because of the relatively high frequency of these three *cre* in their promoters. Individual MYB-related *cre* were present in the analyzed gene promoters at a slightly lower frequency than some of the other *cre*. However, when they were combined into one group, the MYB *cre* frequency was very high (i.e., 8.05, 7.77, and 5.75 per 100 nt for the DEGs, PAGs, and reference genes, respectively). The MYB transcription factors regulate pollen development mainly by modulating the expression of downstream target genes that control tapetum development, programmed cell death, uninucleate microspore differentiation, pollen spore pigment synthesis, mature pollen functions, anther dehiscence, and angiosperm pollen development [80]. The diversity in the effects of these transcription factors may help to explain the high frequency of these *cre*.

Along with the availability of a wheat reference genome [81], molecular markers developed for decades by wheat researchers can now be physically mapped and compared with the positions of candidate DEGs or PAGs with defined expression patterns. To date, several types of molecular markers, including restriction fragment length polymorphism (RFLP), simple sequence repeat (SSR), SNP, kompetitive allele-specific PCR (KASP), and diversity arrays technology (DArT) markers, have been linked to the following seven *Rf* genes: *Rf1* [46,47,82], *Rf2* [50], *Rf3* [39,40,41,42,47,49,82,83], *Rf4* [41], *Rf6* [35], *Rf8* [36], and *Rf9* [39]. The DNA sequences of all available genetic markers flanking the genes and QTLs related to fertility restoration were used to delimit target regions. The *Rf1*, *Rf4*, *Rf8*, and *Rf9* physical regions have been consistently reported [36,39,41,46,49,50,51,52]. The number of candidate genes encoding proteins with PPR domains was determined by URGI. The *Rf8* region on 2D (13.7–23.2 Mb) lacks PPR-encoding genes in the CS reference genome (Appendix A).

Only one anchor marker has been identified for *Rf2* [50]. Additionally, analyses of *Rf6* have produced inconsistent findings. In one earlier study involving RFLP markers, *Rf6* was mapped to the 371.5–445.1 cM region on 5D that includes eight PPR genes annotated in the CS reference genome [41]. In contrast, SSR markers for *Rf6* translocated with the 6U segment from *Ae. umbellulata* to chromosome 6A [84]. A BLAST analysis indicated *Rf6* was localized in the 413.0–437.5 Mb region on 6B, which includes a single annotated PPR gene. The *Rf3* gene has been extensively investigated. We detected the following three regions on 1B that may affect fertility restoration: 16.8–44.07 Mb (*Rf3* explaining 7%–21% of the phenotypic variation), 336.3–363.1 Mb (*QRf.Sv-1B*), and 506.3–557.1 Mb (explaining 16.8%–38.2% of the phenotypic variation) (Appendix A).

Identifying the DEGs and PAGs in physical regions previously confirmed to contain fertility-restoring genes is the most straightforward way to select *Rf* candidate genes. This approach successfully detected *Rf1* and *Rf3* candidate genes. We determined that the *Rf1* region on 1A refers to the 5.0–28.6 Mbp fragment [39]. The SNP marker AX-94682405 positioned at 14.1 Mb is closely linked to *Rf1* [46]. The physical positions corresponding to SNP markers (cfn0522096 and cfn0527067) and *Rf1* were more precisely determined in a recent study by Melonek et al. [47]. Three PPR genes (*TaPPR1–3*) were annotated in this region, one of which was localized to the same position (14.8 Mbp) as DEG1 (RFL79). In our study, DEG1 expression was lower in Patras than in Astoria and Grana in the M stage, whereas the other PPR genes in this region were not differentially expressed.

The mining of the available mapping results for fertility-restoring genes on chromosome 1B generated interesting results. Regarding the mapping of *Rf3*, three physical locations on chromosome 1B affecting fertility restoration were identified. The *Rf3* gene has been most frequently mapped in the 16.8–44.1 Mbp region [39,40,46,49]. However, two additional loci in the 336.2–363.3 Mbp region [49] corresponding to *QRf.Sv-1B* and in the 506.3–557.1 Mbp region have also been reported [49,50]. Ten PPR genes (*TaPPR89–98*) were annotated in the main region of *Rf3*, localized on 1BS, and four DEGs (DEG3, DEG4, DEG5, and DEG6) were mapped in the main locus (18.0–18.8 Mbp). The expression patterns of these DEGs varied, but the expression levels of all four DEGs were lower in Primépi than in Astoria in the T stage. Recent data confirmed that DEG5 (RFL29b) corresponds to *Rf3* [47]. Both DEG3 (RFL164) and DEG4 (RFL396) produced truncated transcripts and DEG6 (RFL58) was not rejected. Interestingly, none of the DEGs or PAGs mapped to chromosome 1BS were overexpressed in Primépi, which is a known *Rf3* donor [31,40,43]. Instead, DEG4 was overexpressed (in both assays) in the fertility-restorer cultivar Patras, which may be considered as a source of *Rf* genes, including known (*Rf3*) and unknown genes located on chromosomes 2A, 2B, 3B, 4A, 5B, 6B, 7A, and 7D.

Two additional fertility-restoring loci were identified on chromosome 1B. The *QRf.Sv-1B* locus, which was responsible for 13.9%–17.6% of the seed set variation, was delimited to the 336.21–363.15 Mb region. Three PPR genes, including PAG34, were mapped in this region. Five PPR genes and one gene encoding a protein with the ADD were mapped (506.33–557.12 Mb) in the second main fertility restoration locus on 1B, which was also reported as *Rf3* [49,50], but no functional markers matched this region.

Two PAGs (PAG17 and PAG16) that were more highly expressed in Patras than in both non-fertility-restoring cultivars in the M and T stages were mapped in the CS reference genome in regions corresponding to *Rf4* and *Rf9*. The *Rf4* region spans a 28-Mb fragment on the short arm of chromosome 6B and contains a cluster of 11 PPR genes, including PAG17, which may be a functional *Rf4* candidate gene. Similarly, *Rf9* was delimited to the 1–23.7 Mb region on 6A, in which 11 PPR genes were mapped, including PAG16.

In the present study, 11 DEGs were mapped on the chromosomes of homoeologous groups 1 and 6, including five DEGs located in regions corresponding to *Rf1* and *Rf3*. The positions of the remaining DEGs on chromosomes of homoeologous groups 2 and 7 were not within known *Rf* loci [39,49,51]. Four DEGs (DEG8–DEG11) assigned to a wheat homoeologous group 2 chromosome belong to the PPR family, and the remaining six DEGs (DEG16–DEG21) on chromosomes 7A, 7B, and 7D are members of the ADD family (Appendix A).

A total of 11 DEGs and 11 PAGs were selected for a qRT-PCR assay to validate the results of the RNA-seq and PAG analyses. The qRT-PCR assay has been broadly used to validate the results of transcriptome-based investigations (e.g., RNA-seq) and is considered to be an independent and highly sensitive tool for targeted analyses. However, an advantage of RNA-seq is that it does not require preliminary knowledge or assumptions about the genes of interest. Thus, it has been successfully used in transcriptomic studies (e.g., [85,86,87]). In some cases, RNA-seq has been applied to validate data [88,89]. Recently, Garriado et al. [90] used qRT-PCR to validate RNA-seq data in their study aimed at identifying wheat reference genes related to meiosis.

Among the genes identified in this study, 10 DEGs and 10 PAGs were chosen because their RNA-seq expression profiles were as expected (i.e., expression level that was higher in at least one fertility-restorer than in at least one non-restorer in at least one anther developmental stage). Both DEG6 and PAG34 were selected because their expression profiles were inconsistent with what was expected (i.e., higher expression level in Astoria than in the fertility-restoring cultivars in the T stage or in both the T and M stages). Furthermore, all DEGs and PAGs included in the qRT-PCR assay encoded the PPR domain, which is the most typical domain encoded by *Rf* genes. Unexpectedly, only some of the DEGs and PAGs had expression profiles that were consistent between the two applied approaches.

The DEG expression data were highly consistent between the in silico and qRT-PCR analyses, whereas relatively consistent expression profiles were obtained for only three PAGs. These results may be related to the specificity and degree of accuracy of both methods. The discrepancies between the gene expression levels determined by RNA-seq and qRT-PCR may also be due to the sequence selected for designing primers (i.e., insufficient specificity). Another possible explanation for the differences between the RNA-seq and qRT-PCR expression patterns involves how extensively the gene families were investigated. Furthermore, alleles may differ regarding their expression levels. There were more than 1000 possible sequences for designing primers for each DEG. In this study, we used universal primers for some DEGs, whereas primers specific to the fertility-restorers were used for other DEGs. In some cases, both primer types were used. The high allelic variation reported for candidate fertility restoration genes leads to variable templates and hinders the designing of gene-specific primers.

The most surprising result was the differential expression of DEG9 and DEG14 in the fertility-restoring cultivars, depending on the analytical method. More specifically, the RNA-seq analysis indicated these genes were overexpressed in Primépi, but the qRT-PCR data suggested both genes were overexpressed in Patras. A similar discrepancy in expression levels was observed for PAG22 and PAG27. These four genes will need to be investigated more thoroughly in future studies.

A total of 14 DEGs and 48 PAGs initially determined to encode the PPR domain were subjected to a deeper structural analysis, which revealed that only eight DEGs and seven PAGs fully satisfied the criteria for identifying genes encoding P-class RFL proteins. Both DEG8 and PAG15 encoded PPR motifs consisting of 14 P-sites (i.e., only one fewer than the generally recognized lower limit). Interestingly, among the RFL-encoding DEGs, DEG1 and DEG2 did not produce the expected expression profiles. A similar result was obtained for PAG5 (i.e., the expression level was lower in the fertility-restoring cultivars than in the non-fertility-restoring cultivars) and for PAG16 (i.e., the expression profile was not verified by qRT-PCR). Moreover, DEG10, DEG12, and PAG27, which were overexpressed in Patras in both analyses, are not functional *Rf* genes because they encode fewer than 15 P motifs.

Twenty-one of the PAGs were designated as genes encoding PLS-class proteins, which are strongly associated with RNA editing (i.e., the post-transcriptional alteration of RNA sequences) [91]. Surprisingly, one of the PLS-encoding PAGs, namely PAG31, was apparently overexpressed in the fertility-restoring cultivar, Patras.

On the basis of our study results, we speculate that seven genes (three DEGs and four PAGs) are *Rf* candidate genes (Table 3). These candidates meet the following criteria: an expected expression pattern (i.e., significantly higher expression level in one or both fertility restorers than in both non-restorers in at least one developmental stage); an expression pattern that was validated by qRT-PCR; the presence of all or most *cre*; and the inclusion of a sequence encoding the PPR domain, indicative of a function related to fertility restoration. In addition to the identification of DEG4 as an *Rf3* candidate gene, six other genes were proposed as candidates for major restorer genes. All selected candidates belong to the PPR family. This set did not include DEG1 mapped in the *Rf1* region or PAG16 and PAG17 mapped in the *Rf9* and *Rf4* regions, respectively, because their expression profiles in both assays were the opposite to what was expected. Together with the genes that were excluded on the basis of the “wet” expressional analysis, the above-mentioned DEGs and PAGs will need to be investigated further using new, redesigned primers. Among the candidate genes, only DEG4 is precisely located in the previously determined position for *Rf3* [39,40,41,42,49,50,82,83]. Surprisingly, DEG4 was clearly overexpressed in Patras and not in Primépi, which is an *Rf3* donor [31,40,43]. The remaining two DEGs and all PAGs were mapped in entirely new positions. Moreover, PAG52 was assigned to chromosome 7B, where neither *Rf* nor *Rf*-related QTLs have been mapped to date. However, no DEGs or PAGs were mapped in the previously determined positions on chromosomes 2DS (with *Rf8*), 6D (with *Rf5*), and 7D (with *Rf2*). It is likely that neither of the two fertility-restoring cultivars included in this study is a source of *Rf2*, *Rf5*, or *Rf8*.

In this study, we proved that an integrated strategy employing RNA-seq, a PAG analysis, and qRT-PCR could efficiently identify new *Rf* candidate genes and evaluate known *Rf* genes, which may be relevant for wheat hybrid breeding. These genes can be used as a source for the development of molecular markers for marker-assisted selection of hybrid paternal components or to create a more sophisticated selection diagnostics based on the analysis of their expression profiles. Both fertility-restoring cultivars were verified as *Rf* donors, although Patras—a variety in which chosen Rf candidates were clearly induced, may be the better donor. This cultivar represents a new source of fertility-restoring genes.

## 4. Materials and Methods

### 4.1. Plant Growth Conditions and Sampling

This study was conducted using two non-restorer commercial cultivars of hexaploid winter wheat, Astoria (Polish cultivar, pedigree: TAW 119452/82 × Elena) and Grana (Polish cultivar, pedigree: Étoile de Choisy × Wysokolitewka Sztywnosłoma × Dańkowska Biała), as well as two cultivars carrying *Rf* genes, Patras (German cultivar, pedigree: Paroli/Toras; seeds obtained from Professor Adam J. Lukaszewski) and Primépi (French cultivar, a source of the *Rf3* gene, pedigree: selected from Superhatif, Yeoman//Barbu-D-Ukraine/Bon-Fermier). Patras can effectively restore the fertility of two cultivars with *T. timopheevii* cytoplasm (Chris_cms_ and Selkrik_cms_) (Professor A. J. Lukaszewski, personal communication). Additionally, its fertility-restoring properties were determined on the basis of field experiments performed over several years by researchers in the Department of Plant Genetics, Breeding and Biotechnology, Warsaw University of Life Sciences.

Plants, after undergoing a 6-week vernalization at 4 °C, were transferred to a growth chamber (ForClean, ZalMed Sp. z.o.o, Warsaw, Poland) and incubated under the following conditions: 14-h light (18 °C)/10-h dark (15 °C) and 50% relative humidity. Plants were cultivated in pots (19 cm diameter) containing a mixture of peat and perlite (1:1, *v*/*v*), with three plants per pot. The plants were hand irrigated daily to ensure the soil water content was close to field capacity. After ear emergence was considered to be complete (i.e., 59 on the Zadoks growth scale), anthers were harvested from spikes at the tetrad (T) and late uninucleate microspore (M) stages. The experimentally determined auricle distance (AD) was used for predicting the anther developmental stage in all four varieties (Appendix A). The RNA-seq analysis was completed using three biological replicates, each comprising 70–80 anthers. The collected anthers were immediately submerged in RNAlater™ Stabilization Solution (Ambion, Thermo Fisher Scientific, Waltham, MA, USA) and then stored at −80 °C.

### 4.2. RNA Isolation

Total RNA was extracted from the frozen anthers using the GeneMATRIX Universal RNA Purification Kit (EURx, Gdańsk, Poland). The quantity and purity of each RNA sample were determined according to the absorbance at 260 and 280 nm, which was measured using the Nanodrop 2000 spectrophotometer and the Qubit^®^ 2.0 Fluorometer (Invitrogen, Waltham, MA, USA). The RNA integrity number for the total RNA was determined using the 2100 Bioanalyzer and the Plant RNA Pico or RNA 6000 Nano Kit (Agilent Technologies, Santa Clara, CA, USA). The RNA was also examined by 1% agarose gel electrophoresis.

### 4.3. RNA-Seq Analysis

The total RNA extracted from the harvested anthers was subjected to a poly(A) enrichment step using the NEBNext Poly(A) mRNA Magnetic Isolation Module (New England Biolabs, Ipswich, MA, USA). Twenty-four strand-specific cDNA libraries (i.e., three biological replicates of the four cultivars at two anther developmental stages) were constructed using the NEBNext^®^ Ultra™ II Directional RNA Library Prep Kit for Illumina^®^ (New England Biolabs, Ipswich, MA, USA). The libraries were sequenced using the HiSeq 4000 system (Illumina) in the PE150 mode, and the data were analyzed by Genomed S.A.

Regarding the data analysis, the Cutadapt program was used in the paired-end mode to remove adapters (only from the 3′ side) and polyA sequences (and analogously generated tails with the polyT sequence). Short reads (<36 bp) were also eliminated. The retained clean reads were mapped to the *T. aestivum* reference genome (GenBank accession no. GCA_900519105.1, NCBI, Bethesda, MD, USA) using TopHat [92]. More specifically, the TopHat options appropriate for the fr-firststrand library were used in the no-novel-juncs mode. Next, the number of reads mapped to individual genes was calculated using the HTseq program [93] (Anders et al., 2014), with transcript strand differentiation (–stranded = reverse). The mapping data were processed in the R environment using the DESeq2 package to identify differentially expressed genes (DEGs) according to the following criteria: threshold false discovery rate of 0.05 and an absolute log_2_(fold-change) ≥0.5. The DEGs were identified by comparing the non-restorers and restorers individually and in pairs at the T and M stages. Sequencing data were deposited in the *ArrayExpress* database via *Annotare* (https://www.ebi.ac.uk/arrayexpress accessed on 16 August 2021).

The DEGs were functionally characterized on the basis of gene ontology (GO) terms, which were assigned using the biomaRt and topGO package [94]. A Venn diagram was constructed using the jvenn tool [95]. Amino acid sequences encoded by the *Rf* genes from Poaceae species and the consensus PPR motif retrieved from the NCBI database and from a published article [96] were incorporated into a local database and used for selecting DEGs (Appendix A). The DEGs related to the sequences in the local database were filtered out in the R environment using the BLAST program. To identify potential *cis*-regulatory elements (*cre*) in the promotor sequences of the selected DEGs and reference genes, the sequences 2000 bp upstream of the genes were screened using PlantCare [97].

### 4.4. Paralog Analysis

The Ensembl Plants database (http://plants.ensembl.org/ accessed on 16 August 2021) was used to search for the PAGs of 21 selected DEGs. The PAGs with a mean expression level (i.e., mean number of mapped reads) in the samples exceeding 10 were clustered according to an unsigned weighted correlation network analysis in the R package WGCNA, with the following parameters: beta = 1; clustering method “complete”; cutting method “cutreeStaticColor”; cutHeight = 0.40; minSize = 10 [98,99].

### 4.5. Mapping DEGs and PAGs in Rf Linkage Groups

On the basis of structural comparisons, the DEGs and PAGs were mapped to previously reported chromosomal segments containing fertility-restoring genes. Markers linked to the fertility restoration genes *Rf1*–*Rf4*, *Rf6*, *Rf8*, and *Rf9* along with QTLs were retrieved from published reports [36,39,40,41,46,47,50,52]. Next, the DNA sequences of the respective RFLP probes, SSR markers (or primers), and SNP markers were retrieved from CerealsDB (https://www.cerealsdb.uk.net/cerealgenomics/CerealsDB/indexNEW.php accessed on 16 August 2021) or GrainGenes (https://wheat.pw.usda.gov/GG3/ accessed on 16 August 2021). The sequences served as queries for a BLAST search of the Chinese Spring (CS) wheat genome assembly (IWGSC RefSeq v1.0) by URGI (https://urgi.versailles.inrae.fr/blast/ accessed on 16 August 2021). The best hits for the target chromosomal regions were selected to call marker physical positions. Similarly, the positions of all DEGs selected in this study on the reference map were determined. Additionally, the positions of all PPR genes in the target regions were determined by URGI and added to the partial physical maps of the chromosomal segments with mapped fertility-restoring genes. 

### 4.6. Validation of Transcripts by RT-qPCR Analysis

The relative expression levels of genes selected according to the RNA-seq and PAG analyses were validated by qRT-PCR (Appendix A). Total RNA (500 ng) served as the template for synthesizing the first-strand cDNA using the smART First Strand cDNA Synthesis Kit (EurX) and an oligo(dT)_20_ primer. The qRT-PCR assay was performed using a 48-well plate and the Eco Real-Time PCR System (Illumina, San Diego, CA, USA). The 10-µL qRT-PCR mixture included 2 µL cDNA, 500 nM of each gene-specific primer, and 5 µL 2× GoTaq qPCR Master Mix (Promega, Madison, WI, USA). The qRT-PCR analysis was performed using three biological replicates and two technical replicates. The PCR program was as follows: 95 °C for 2 min; 40 cycles of 95 °C for 15 s, and 60 °C for 60 s. The gene-specific primers were designed according to the cDNA sequence fragments conserved across Grana, Astoria, Patras, and Primépi (universal) or the sequence fragments useful for distinguishing between fertility-restoring and non-fertility-restoring cultivars, which were determined from the RNA-seq clean reads (Appendix A). The *HvAct* (GenBank accession no. AY145451) and cyclic phosphodiesterase (CPD)-like (GenBank accession no. AK453202.1) genes [90] were used as internal reference controls. The expression levels of the analyzed genes were normalized against the internal reference gene expression levels using the 2^−ΔΔCt^ method [100]. The significance of the differences between ΔCt values was assessed by the Mann–Whitney U test (with continuity correction) in the STATISTICA 12 package (TIBCO Software, Palo Alto, CA, USA).

### 4.7. Annotation of PPR Genes

The PPR domains of the selected DEGs and PAGs (Appendix A) were annotated by Plant Energy Biology [91]. The restorer-of-fertility-like (RFL) proteins among the P-class PPR subfamily members were selected on the basis of the presence of tandem arrays comprising 15–20 PPR motifs, each composed of 35 amino acid residues [47].

## Figures and Tables

**Figure 1 ijms-22-09146-f001:**
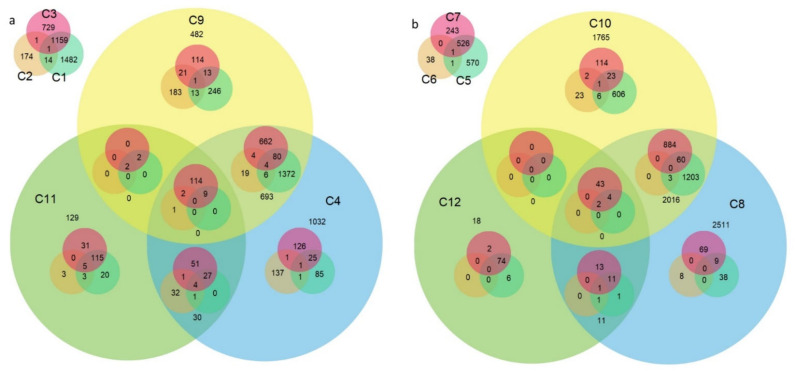
Nested Venn diagrams of the number of DEGs at the T (**a**) and M (**b**) microspore developmental stages. Within the T and M stages, the colored circles refer to unique comparisons. C1, C5: Astoria/Primépi; C2, C6: Grana/Primépi; C3, C7: Astoria/Patras; C4, C8: Grana/ Patras; C9, C10: Patras/Primépi; and C11, C12: non-restoring/restoring cultivars (Table 1).

**Figure 2 ijms-22-09146-f002:**
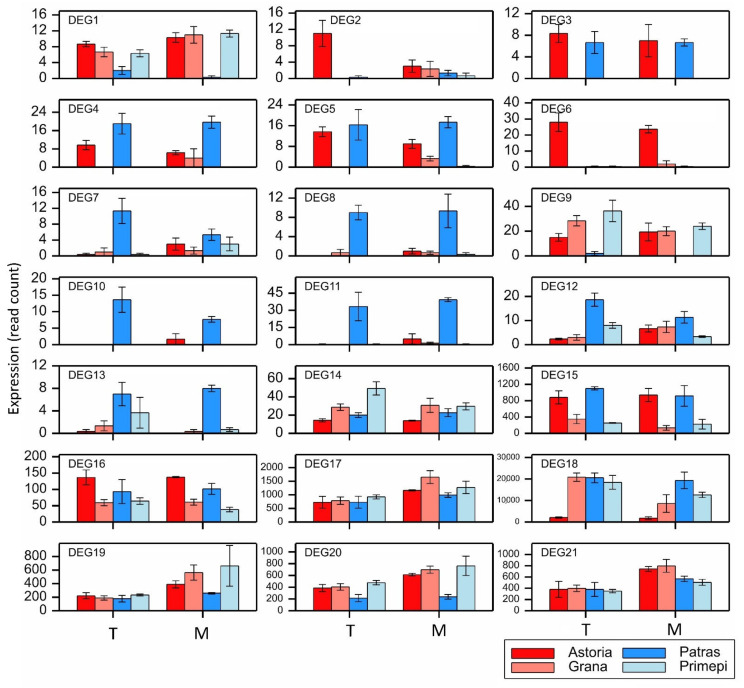
In silico expression profiles of 21 DEGs identified as potential *Rf* candidate genes. Data are presented as the mean values of the NGS read counts over the replicates, with error bars corresponding to the standard error of the mean.

**Figure 3 ijms-22-09146-f003:**
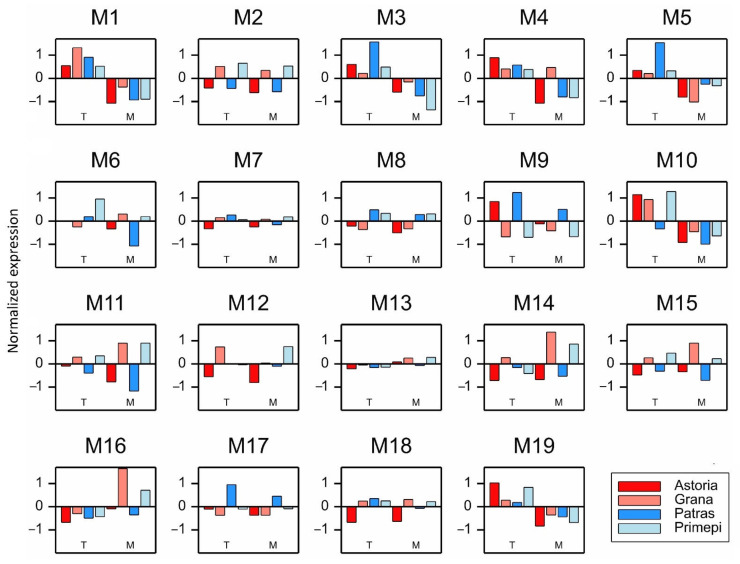
Average expression profiles of genes in 19 clusters. Data are presented as average normalized values in WGCNA.

**Figure 4 ijms-22-09146-f004:**
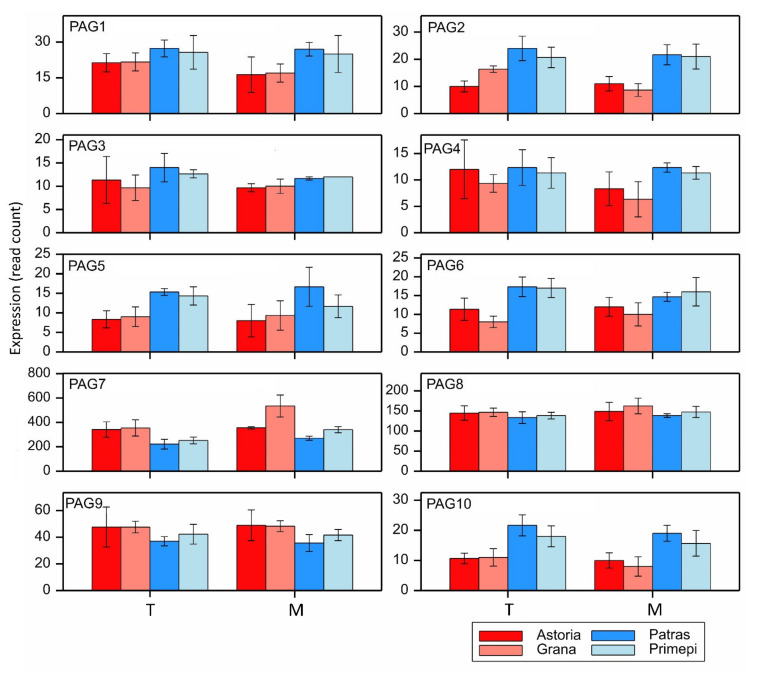
Expression patterns of 10 PAGs in cluster M8. Data are presented as the mean values of the NGS-read counts over the replicates, with error bars corresponding to the standard error of the mean.

**Figure 5 ijms-22-09146-f005:**
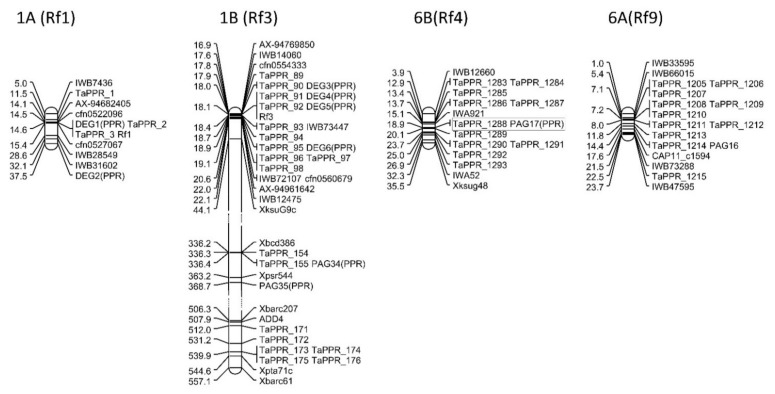
Physical maps of the *Rf1*, *Rf3*, *Rf4*, and *Rf9* loci with co-localized DEGs and PAGs. Marker positions are provided in Mbp. TaPPR markers refer to annotated PPR genes (URGI). The DEGs and PAGs that were localized within any *Rf* locus are framed.

**Figure 6 ijms-22-09146-f006:**
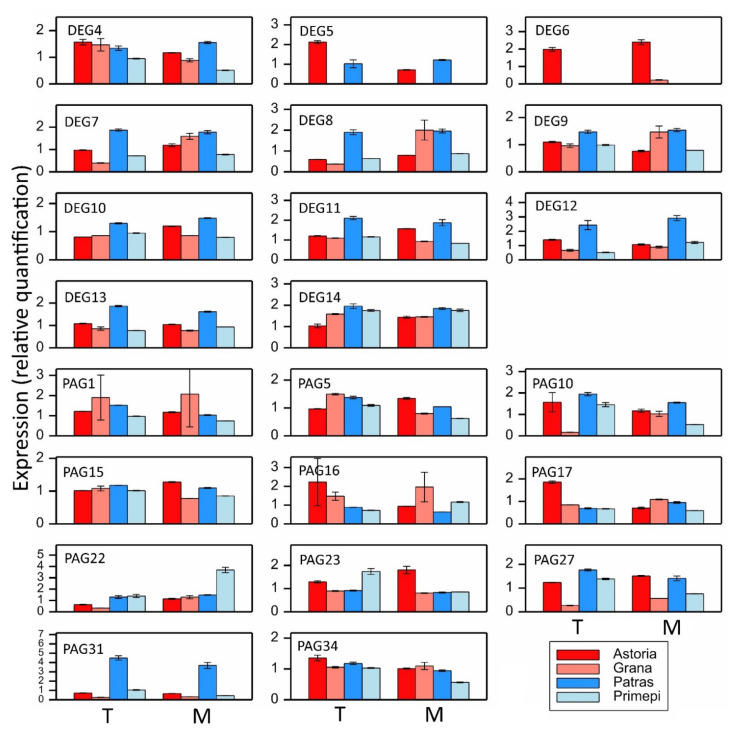
Expression profiles of selected DEGs and PAGs as determined by qRT-PCR. Data are presented as relative quantification with standard error bars.

**Table 1 ijms-22-09146-t001:** Number of upregulated and downregulated * DEGs identified in the RNA-seq analysis.

Comparison No.	Stage and Compared Cultivars	Number of DEGs	
		Upregulated	Downregulated
C1	T: Astoria/Primépi	2372	2317
C2	T: Grana/Primépi	43	590
C3	T: Astoria/Patras	1729	1574
C4	T: Grana/ Patras	2188	2330
C5	M: Astoria/Primépi	1555	1592
C6	M: Grana/Primépi	21	66
C7	M: Astoria/Patras	951	1131
C8	M: Grana/Patras	3014	3874
C9	T: Patras/Primépi	1912	2129
C10	M: Patras/Primépi	3520	3235
C11	T: non-restoring/restoring	287	295
C12	M: non-restoring/restoring	87	100

* Relative to the first cultivar in each comparison. Total RNA was isolated from anthers collected from the non-restoring (Astoria and Grana) and fertility-restoring (Primépi and Patras) cultivars at the tetrad (T) and late uninucleate microspore (M) stages.

**Table 2 ijms-22-09146-t002:** Characteristics of 21 DEGs selected by a targeted RNA-seq analysis using the Ensembl Plants database.

DEG	Gene Symbol	No. of Exons	Variant Alleles	Comparison	Domain *	Reference Sequence with the Highest Similarity **	Score (%)	E-Value
DEG1	*TraesCS1A02G031600*	1	1	C7 C8	PPR	*BrachypodiumRf1*	59.19	0.0
DEG2	*TraesCS1A02G057400*	3	138	C1 C3	PPR	*Ae. taushii Rf1*	62.61	0.0
DEG3	*TraesCS1B02G038300*	2	266	C1 C4 C8	PPR	*BrachypodiumRf1*	57.24	2e-159
DEG4	*TraesCS1B02G038400*	2	85	C1 C4 C9 C10	PPR	*BrachypodiumRf1*	61.07	0.0
DEG5	*TraesCS1B02G038500*	2	160	C1 C4 C8	PPR	*BrachypodiumRf1*	59.95	0.0
DEG6	*TraesCS1B02G039200*	1	238	C1 C3 C5 C7	PPR	*BrachypodiumRf1*	59.69	0.0
DEG7	*TraesCS1B02G075000*	1	20	C3 C9	PPR	*Ae. taushii Rf1*	76.38	0.0
DEG8	*TraesCS2A02G530300*	2	60	C3	PPR	*PPR-814a*	54.18	0.0
DEG9	*TraesCS2A02G530600*	1	52	C4 C7 C8	PPR	*PPR-814a*	57.66	0.0
DEG10	*TraesCS2B02G560000*	1	132	C3 C4 C8	PPR	*Ae. taushii Rf1*	55.64	3e-174
DEG11	*TraesCS2B02G560700*	1	239	C3 C4 C8	PPR	*BrachypodiumRf1*	56.76	0.0
DEG12	*TraesCS6A02G099318*	1	7	C3 C4	PPR	*BrachypodiumRf1*	57.75	7e-106
DEG13	*TraesCS6A02G099384*	1	9	C7 C8	PPR	*PPR-814a*	49.18	1e-38
DEG14	*TraesCS6B02G129200*	2	208	C1	PPR	*BrachypodiumRf1*	58.26	0.0
DEG15	*TraesCS6B02G321400*	7	38	C1 C4 C5 C8 C9 C10	ADD	*ZmRf2*	76.22	0.0
DEG16	*TraesCS7A02G369300*	10	285	C5	ADD	*ZmRf2*	55.67	0.0
DEG17	*TraesCS7A02G369400*	9	250	C8	ADD	*ZmRf2*	55.05	0.0
DEG18	*TraesCS7B02G116800*	11	290	C1 C3 C5 C7	ADD	*ZmRf2*	87.39	0.0
DEG19	*TraesCS7B02G259100*	8	222	C8	ADD	*ZmRf2*	56.01	0.0
DEG20	*TraesCS7D02G353900*	8	196	C7 C8	ADD	*ZmRf2*	55.26	0.0
DEG21	*TraesCS7D02G354000*	9	201	C12	ADD	*ZmRf2*	55.26	0.0

*Aegilops tauschii Rf1* accession number: XP_020153657.1; *Brachypodium distachyon Rf1* accession number: XP_024315805.1; *PPR-814a* accession number: ACN24620.1; *ZmRf2* accession number: NP_001105891.1. * Domains: ADD: aldehyde dehydrogenase domain; PPR: pentatricopeptide repeat domain. ** The number and identity of the reference sequences highly similar (based on the E-value) to the DEGs are provided in Appendix A; the wheat genes encoding the ADD are listed in Appendix A.

**Table 3 ijms-22-09146-t003:** Probable *Rf* candidate genes among the DEGs and PAGs.

DEG/ PAG	Chromo-Some	Position in CS Genome (bp)	Class/ Domain	Missing *cre* Indicating *Rf* Function	Candidate for the Gene:	Overexpression Pattern *
DEG4	1BS	18093637–18090927	PPR	none	*Rf3 ***	M: Pa > A,G
DEG7	1B	58116320–58114326	PPR	ARE, MYB	*Rf_a ****	T,M: Pa > A,G; T: Pr > G
DEG11	2B	753931307–753928992	PPR	ARE, CGTCA-motif, TGACG	*Rf_b* (*RFL_2018_*) ****	T,M: Pa > A,G
PAG1	1B	56620540–56618322	PPR	ARE	*Rf_c ****	T:Pa > A,G; M: Pa > G; T: Pr > G
PAG10	5B	36720490–36722627	PPR	none	*Rf_d ****	T,M: Pa > A,G; T: Pr > G
PAG25 *****	5A	546692034–546697035	PPR	ARE	*Rf_e****	T,M: Pa > A,G; T,M: Pr > A,G
PAG52 *****	7B	3846939–3849534	PPR	none	*Rf_f****	M: Pa > A,T; T:Pr > A; M: Pr > A,G

* phases: T—tetrad stage, M—microspore stage; genotypes: Pa—Patras, Pr—Primepi, As—Astoria, Gr—Grana ** precise localization determined previously for *Rf3* [39,40,41,42,49,50,82,83]. **** Rf_a–Rf_f* are newly identified *Rf* candidate genes. **** RFL_2018_ is a restorer-of-fertility-like gene mapped by IWGC [81]. ***** not verified by qRT-PCR.

## Data Availability

Data available in a publicly accessible repository. The data presented in this study are openly available in https://www.ebi.ac.uk/arrayexpress/experiments/E-MTAB-10131/ accessed on 16 August 2021 at reference number PRJEB43230.

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
