# Peer review of "Identification of Rf Genes in Hexaploid Wheat (Triticumaestivum L.) by RNA-Seq and Paralog Analyses"

_ijms, 2021, doi:10.3390/ijms22179146_

Round 1

Reviewer 1 Report

The manuscript entitled "Identification of Rf genes in hexaploid wheat (Triticum aestivum L.) by RNA-seq and paralog analyses" has the aim to describe a novel bioinformatic approach to find new fertility-restoring (Rf) genes in bread wheat but failed to do that because the story is not told in a clear / easily understandable way.

Anyway, the MS designated seven Rf candidate genes that could help in breeding programs and, if improved, could deserve publication.

However, there are a number of inaccuracies and points that must be better explained to make easier the comprehension of this manuscript.

  • In Introduction it is necessary to indicate which are the two fertility restoring and non restoring cultivars (it seem that is only indicated in the bottom of Table 1)
  • The selection of the genotypes should be better explained. Generally, in transcriptomic analysis, the genetic background of the compared genotypes should be as similar as possible. The authors selected Astoria and Grana as not restored genotypes and Primepi and Patras as restored genotypes. They look very different, so the 20,000 identified DEG are probably due to genotypic differences among these genotypes.
  • In the Result section it is necessary briefly explain the ratio of some choices or analyses (e.g. why the T and M stages or why go to the next step)
  • Table 1: the C9 and C10 comparisons are different from the other comparisons being between two cultivars at the same time so that they are more selective; that is no considered and, any case, C9 and C10 must be the final comparisons no. C11 and C12
  • Figure 1: a) nowhere is it written that a Venn diagram must be made with circles or ovals, however this is the choice that best enables understanding; b) the meaning of the different colours is not specified; c) the bottom part of the figure it is not understandable and the poor legend does not help at all
  • Table 2: at line 164-165 it is indicated that only genes with a sequence similarity and sequence coverage exceeding 50% were selected but in the table, for gene TraesCS6A02G099384 [DEG13] the value is 49.18; moreover, being used the term “DEGx” in the text “DEGx” must be the main indication in the first column, not in brackets
  • Figure 2: a) DEG1, 2, 3, 7, 8 and 13 have very low expression levels compared to the other genes. Please discuss it, in particular in comparison to qRT-PCR validation data; b) please omit the gene numbers; c) please use e.g. two shades of red for the two non-restoring cultivars and two shades of blue for the two restoring cultivars
  • Paragraph 2.1.3 and figure 3. Because the 21 DEGs should be Rf genes the GO analysis is unnecessary, as well as figure 3.
  • Figure 4 and Figure 5. Figure 4 shows clusters (why did you call clusters with colours names? You can call them cluster 1, 2...19) and the y-axis report a median-normalized expression level, but subsequently, in figure 5 the authors show the PAGs of the "Light Cyan" cluster using a different y-axis scale. Please unify the y-axis of figure 4 and 5 using the same scale, number cluster from 1 to 19 and use in Figure 5 DEG or PAG codes.
  • Figure 7 please add in each frame the corresponding DEG or PAG, and change the colors for the cultivars
  • Table 3: it is necessary to explain the meaning of the abbreviations in the “Overexpression pattern” column and their relevance
  • Finally, please discuss how the 7 Rf candidate genes could be employed for developing new hybrid cultivars, and why Patras should be a better donor.

Reviewer 2 Report

The present manuscript presents a very interesting subject. It a well written manuscript which I enjoyed reading. The English needs a little "polishing" though. The research design was carefull orchestrated/conducted and the conclusions are well supported by the results.  I see no major mistakes and the manuscript could be published after a minor revision. The introduction and the discussion parts are a bit long and could be shorten, while the quality of the figures is also low. 

Author Response

Thank you very much for positive opinion. We have improved quality of figures. Manuscript was oryginally corrected by Edanz.

Round 2

Reviewer 1 Report

The Authors answered to all the points raised in the revision. Anyway, I have one more suggestion: for Figure 1 I prefer the Edward’s Venn as showed in the coverletter with each comparison number in bold and of the same color of the respective border line.